# Agentic Science: A Self-Automated Research Paradigm Based on Dynamic Knowledge Graphs and Multi-Agent Systems

## Abstract

Artificial intelligence is fundamentally reshaping the paradigms and methodologies of scientific research.This paper proposes a novel self-automated research paradigm based on dynamic knowledge graphs and multi-agent collaboration, aiming to achieve end-to-end intelligent processing from literature mining to knowledge discovery. The core innovation lies in the integration of large language models' semantic understanding capabilities with knowledge graphs' structured reasoning capabilities, through mechanisms such as multi-stage knowledge extraction, temporal evolution analysis, and semantic disambiguation optimization, to construct a research knowledge system capable of autonomous evolution. To address the challenges of traditional research automation—such as limited knowledge representation and insufficient complex reasoning—this study presents systematic solutions. Validation in the field of Retrieval-Augmented Generation (RAG) demonstrates that the paradigm can automatically identify temporal evolution patterns of research challenges and generate high-fidelity research analyses and development forecasts. This work lays a methodological foundation for "Agentic Science" and drives the intelligent transformation of scientific research paradigms.

## 1 Introduction

Scientific research is facing dual challenges of knowledge explosion and increasing complexity. Traditional research models, highly dependent on individual researchers' cognitive abilities, suffer from efficiency bottlenecks and subjective biases. With the rapid advancement of artificial intelligence, transformative opportunities for research paradigms have emerged.

This paper proposes a new self-automated research paradigm based on dynamic knowledge graphs and multi-agent collaboration. **Self-automated research** refers to AI systems that can independently complete the entire research workflow—from problem identification and literature review to knowledge construction, trend analysis, and report generation—with human experts providing guidance at key decision points, but without requiring continuous human intervention. This paradigm aims to address core challenges in traditional research automation, such as limited knowledge representation and insufficient complex reasoning. Experimental validation shows that the paradigm possesses strong generalization capabilities and can be widely applied across diverse scientific domains.

## 2 Related Work

Research on automated scientific analysis spans multiple fields, including knowledge graph construction, natural language processing, and intelligent agent systems. This section systematically analyzes the progress and limitations of related research.

Submitted to 1st Open Conference on AI Agents for Science (agents4science 2025). Do not distribute.

## 2.1 Challenges of Knowledge Graphs in Scientific Research

Scientific literature contains highly dynamic knowledge, with new concepts and methods continuously emerging, requiring knowledge graphs to possess rapid evolution capabilities. Existing systems such as Temporal Knowledge Graphs primarily focus on simple temporal tagging, lacking deep temporal reasoning. Furthermore, scientific concepts are expressed in diverse ways; the same concept may appear differently across documents, necessitating robust semantic disambiguation mechanisms.

## 2.2 Limitations of Natural Language Processing in Research

Scientific literature typically contains numerous technical terms and complex logical relationships. General pre-trained models perform poorly on such texts. Research questions often require integrating information from multiple papers for complex reasoning, while existing RAG systems lack deep multi-document analysis capabilities.

## 2.3 Bottlenecks in Intelligent Agent Systems for Research

Existing agent systems face three core challenges in research scenarios: lack of domain-specific optimization, making it difficult to handle the specificity of research tasks; imperfect collaboration mechanisms among agents, leading to low execution efficiency; and inadequate human-machine interaction mechanisms, failing to fully leverage expert knowledge.

## 2.4 Systemic Deficiencies in Research Automation

Current research automation systems primarily focus on specific sub-tasks, lacking integrated, end-to-end solutions. Core issues include: (1) lack of structured knowledge representation, hindering complex reasoning; (2) insufficient dynamic evolution capabilities, unable to adapt to rapidly changing research environments; (3) limited cross-domain generalization; and (4) absence of effective human-machine collaboration. The proposed paradigm provides comprehensive solutions to these shortcomings.

# 3 Methodology and Core Technologies

## 3.1 Overall Paradigm Framework

This study proposes a novel self-automated research paradigm based on dynamic knowledge graphs and multi-agent collaboration. The framework consists of three core modules: knowledge construction, intelligent collaboration, and human-machine interaction, reflecting the transformation from data to knowledge to wisdom.

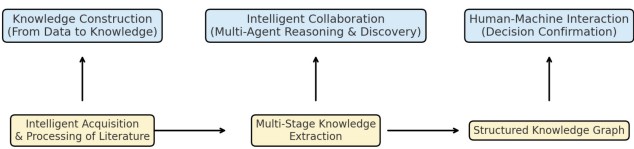

Figure 1: Self-Automated Research Paradigm Concept.

The workflow begins with intelligent acquisition and processing of scientific literature. Using adaptive crawlers and API interfaces, the system collects the latest research papers from academic platforms such as arXiv and PubMed. After preprocessing, raw documents enter a multi-stage knowledge extraction process, building a structured scientific knowledge graph. Based on the knowledge graph, a multi-agent system collaboratively performs analysis, reasoning, and knowledge discovery. Throughout the process, the human-machine interaction module provides confirmation at critical decision points, ensuring the paradigm's reliability.

## 3.2 Dynamic Knowledge Graph Construction Methodology

### 3.2.1 Multi-Stage Knowledge Extraction Process

Knowledge graph construction employs an innovative three-stage process: initial extraction, isolated connection, and skeleton completion. In the initial extraction stage, the system uses predefined entity and relation types to perform structured knowledge extraction via large language models. This stage adopts a fine-grained prompt-based control strategy to ensure high extraction accuracy.

The isolated connection stage handles completely isolated knowledge units (subjects and objects appearing only once in the entire graph). Through statistical analysis and LLM reasoning, the system establishes connections between these isolated units and non-isolated entities. This mechanism significantly reduces knowledge fragmentation.

The skeleton completion stage dynamically generates questions for missing predefined entity types based on paper content, using full-text question answering to fill knowledge gaps. This mechanism ensures the completeness and consistency of the knowledge graph.

### 3.2.2 Temporal Evolution Analysis Mechanism

To address the dynamic nature of scientific knowledge, the paradigm implements multi-dimensional temporal analysis. By using LLMs to automatically identify temporal expressions in questions and convert them into precise time ranges, the system supports multi-granularity time queries and accommodates various date formats.

Based on paper publication timestamps, the system constructs an efficient time index to support rapid time-range queries. Through time-series data mining, the system can identify development trajectories and evolution patterns in scientific fields, providing support for trend analysis.

### 3.2.3 Semantic Disambiguation and Optimization Mechanism

Semantic disambiguation adopts a multi-level matching strategy: exact name matching $\rightarrow$ semantic similarity search $\rightarrow$ LLM intelligent judgment $\rightarrow$ human confirmation. For different entity types, the system defines specificity matching rules. For example, model types focus on parameter scale, while technology types focus on core concept consistency.

When entity similarity falls within a critical range, the system uses LLMs for final judgment to avoid mis-matching. Additionally, the system provides interactive entity merging and confirmation functions, allowing user participation in the disambiguation process.

Context-aware entity optimization uses LLMs to determine whether an entity needs to be verified against the original text, preventing over-optimization. The system identifies overly abstract entity names and provides more precise definitions and descriptions based on original context.

## 3.3 Multi-Hop Knowledge Reasoning System

### 3.3.1 Intelligent Question Decomposition and Intent Recognition

Facing complex research questions, the system employs an LLM-based automatic decomposition mechanism. A complex question $Q$ is decomposed into $n$ logically clear sub-questions $\{q_1, q_2, ..., q_n\}$, maintaining logical order and interdependence. The mathematical model for question decomposition is:

$$Q \rightarrow \{q_1, q_2, ..., q_n\} \quad \text{where} \quad n \leq 5$$

Each sub-question $q_i$ must satisfy the semantic completeness condition:

$$\text{sim}(q_i, Q) \geq \theta, \quad \theta = 0.7$$

where sim is the semantic similarity function, computed using cosine similarity:

$$\text{sim}(a, b) = \frac{a \cdot b}{\|a\| \cdot \|b\|}$$

### 3.3.2 Multi-Hop Reasoning and Path Selection

Multi-hop reasoning adopts an LLM-based intelligent relation selection mechanism. Given entity $e$ and relation set $R$, the probability of selecting the most relevant relation is modeled as:

$$P(r|e, q) = \text{softmax}(\mathbf{w}^\top \phi(e, r, q))$$

where $\phi(e, r, q)$ is the joint feature representation of entity-relation-question, and $\mathbf{w}$ is a learnable parameter.

The path scoring function integrates multiple factors:

$$\text{Score(path)} = \sum_{i=1}^{k} [\alpha \cdot \text{rel\_relevance}(r_i) + \beta \cdot \text{entity\_importance}(e_i) + \gamma \cdot \text{context\_match}(c_i)]$$

where $k$ is the path length, $\alpha + \beta + \gamma = 1$, and $\alpha, \beta, \gamma > 0$.

### 3.3.3 Answer Generation and Confidence Calculation

Answer generation is based on constructing structured evidence chains, with confidence calculation using a multi-factor weighted model that comprehensively considers the number of supporting evidences, semantic coherence, and source reliability, ensuring the credibility and traceability of answers.

## 3.4 Temporal Evolution Analysis Model

To address the dynamic nature of scientific knowledge, the paradigm implements multi-dimensional temporal analysis. Through time-decay weighted models, it handles the time sensitivity of knowledge, and establishes technology trend prediction models based on time-series data, providing quantitative support for research trend tracking.

## 3.5 Multi-Agent Collaboration Framework

The paradigm adopts a multi-agent collaboration framework, simulating the working style of human research teams. Each agent specializes in a specific task type, such as literature retrieval, data analysis, and paper writing. Agents communicate and collaborate through standardized interfaces.

Agent configuration uses a modular design, allowing behavior parameters to be managed via configuration. Toolchain integration enables agents to invoke various functional modules, enhancing the paradigm's flexibility and extensibility. Task decomposition breaks down complex research tasks into parallel-executable sub-tasks, optimally allocating them based on task type and agent capability.

Result integration combines multiple agents' outputs into a consistent final result. The system supports real-time progress feedback and status management, providing a good user experience.

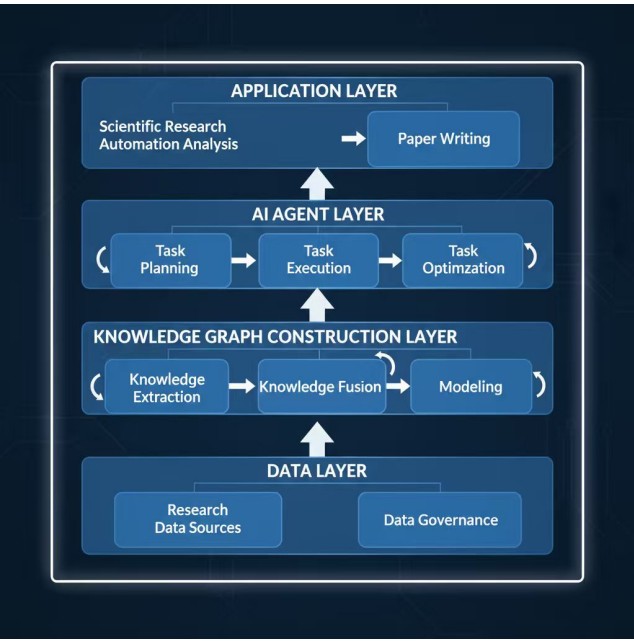

Figure 2: A Multi-Layer Framework for Scientific Research Automation.

## 3.6 User Interaction Mechanism

To meet the special requirements of research scenarios, the paradigm implements multiple types of user interaction confirmation mechanisms, including multiple-choice confirmation (for entity matching and relation selection), question-answer confirmation (for complex decisions), text input (for user-defined input), and file upload (supporting documents and images).

Confirmation triggers are based on confidence and importance assessment. The system intelligently determines whether user confirmation is needed through abstraction level judgment and completeness checks. Threshold mechanisms prevent over-confirmation, balancing accuracy and user experience.

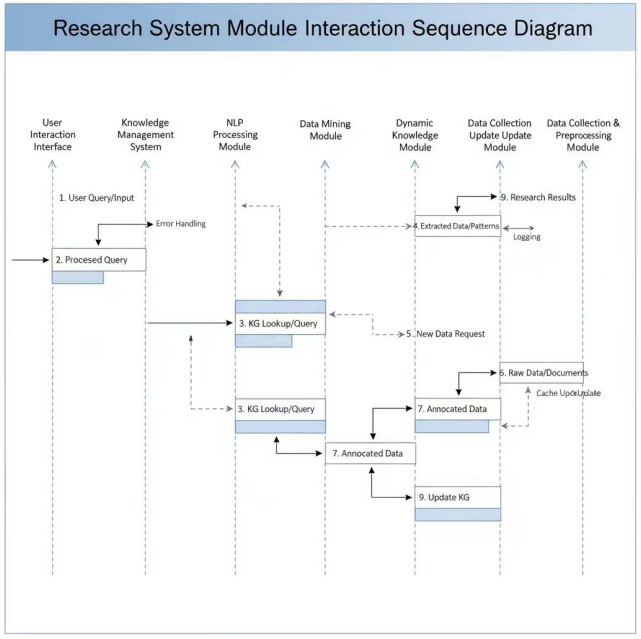

Figure 3: Interaction Workflow of a Knowledge-Driven Research System.

# 4 Experiments and Results Analysis

## 4.1 Experimental Design

The primary goal of this experiment is to empirically validate whether the proposed paradigm can achieve the original intention of **autonomously discovering research hotspots, tracking their evolution, and generating high-quality analysis reports**. The experimental design strictly simulates the workflow of human researchers: **identifying research gaps → formulating key questions → tracking temporal evolution → drawing analytical conclusions**.

The experiment selects "Retrieval-Augmented Generation (RAG)" in the field of artificial intelligence as the target domain, due to its rapid development and dense paper output, making it ideal for testing the system's dynamic analysis capabilities.

The experimental method consists of two phases: 1. **Knowledge Base Construction and Problem Discovery**: The agent autonomously retrieves and crawls papers related to RAG from arXiv during the specified period (January to August 2025). Using the knowledge graph construction tools provided by the paradigm, the system automatically processes the literature and structures the knowledge. Subsequently, the system discovers core research challenges, difficulties, and opportunities in the field through a **multi-source information fusion strategy** (including querying general knowledge sources like Wikipedia, querying the newly constructed domain knowledge graph, and calling third-party Deep Research tools). Finally, the LLM is used to deduplicate, merge, and rank the multi-source discovered problems, forming a representative "core problem set." 2. **Temporal Evolution Analysis**: For each core problem identified above, the agent queries the system at different time points (e.g., "What breakthroughs were achieved in [technique] regarding [difficulty] in [January 2025]?" ) using **time (month) as the independent variable**, obtaining answer slices at different time points. By comparing these answers, the system automatically generates dynamic evolution analysis of each difficulty, including progress speed, breakthrough timing, and current status.

## 4.2 Validation of Autonomous Research Workflow Effectiveness

The experiment successfully validated the full-process automation of intelligent agent-driven research analysis. The system first automatically completed the collection and knowledge processing of RAG-related literature, laying a structured factual foundation for subsequent analysis.

More importantly, the "core problem set" generated by the multi-source strategy accurately captured the key contradictions in the RAG field, such as **information fragmentation in long-context processing, alignment challenges in multimodal retrieval, and efficiency bottlenecks under real-time requirements**. These problems align closely with expert human judgment, proving the effectiveness of the agent in **autonomously defining research problems**.

## 4.3 Dynamic Evolution Analysis Results

This experiment used a multi-model collaborative scoring mechanism (ChatGPT-5, Claude-4, Gemini-2.5) to quantitatively evaluate the technical breakthroughs of nine core difficulties in the RAG field from January to August 2025. The scoring criteria were: 0 points (no technical breakthrough), 0.3 points (slight improvement but not fundamentally solved), 0.5 points (major progress but far from resolution), 0.8 points (major breakthrough with high evaluation metrics), and 1.0 points (complete breakthrough or industry benchmark). The results are shown in Table 1.

### 4.3.1 Temporal Evolution Characteristics of Technical Breakthroughs

The technical breakthroughs in the RAG field exhibit clear phase characteristics. During January and February, most difficulties were in the initial exploration stage; March marked the first breakthrough period; June became a critical turning point, with multiple difficulties making significant progress simultaneously. There are significant synergistic evolution relationships among different technical difficulties. For example, the breakthrough in "context understanding and semantic coherence" is closely related to the subsequent progress in "contextual consistency of generated content," demonstrating the intrinsic logic of technological development. This analysis result shows that the paradigm can effectively identify key nodes and breakthrough patterns in technological evolution, providing deep insights into field development.

Table 1: Timeline assessment of key technical challenges and breakthroughs in the RAG domain

| Challenge description | Jan | Feb | Mar | Apr | May | Jun | Jul | Aug |
|---|---|---|---|---|---|---|---|---|
| Retrieval accuracy: RAG systems struggle to model complex cross-document and cross-source knowledge relationships in multi-hop reasoning tasks; performance on specialized domain questions is poor. | 0.3 | 0.3 | 0.5 | 0.0 | 0.5 | 0.8 | 0.8 | 0.8 |
| Multilingual document handling: challenges in cross-language semantic alignment and consistency. | 0.0 | 0.0 | 0.0 | 0.3 | 0.3 | 0.3 | 0.3 | 0.3 |
| Real-time data update capability: RAG systems commonly face delays in knowledge updates. | 0.0 | 0.0 | 0.3 | 0.3 | 0.3 | 0.3 | 0.3 | 0.5 |
| Contextual consistency of generated content: RAG systems find it difficult to achieve coherent cross-document semantic integration in multi-hop reasoning. | 0.0 | 0.3 | 0.5 | 0.5 | 0.5 | 0.5 | 0.5 | 0.8 |
| Resource consumption and deployment cost: building and maintaining knowledge graphs requires substantial computing resources and data preprocessing costs. | 0.3 | 0.3 | 0.3 | 0.5 | 0.5 | 0.5 | 0.5 | 0.8 |
| Multilingual support: current RAG systems heavily rely on English corpora, lacking unified, high-quality multilingual knowledge graph infrastructure. | 0.0 | 0.0 | 0.0 | 0.0 | 0.0 | 0.0 | 0.0 | 0.0 |
| Credibility and explainability of generated content: RAG systems still have notable shortcomings, with prominent factuality issues. | 0.3 | 0.3 | 0.3 | 0.5 | 0.8 | 0.8 | 0.8 | 0.8 |
| Context understanding and semantic coherence: chunk-based retrieval architectures lack structured semantics, making it hard to handle long-range operations and context forgetting. | 0.0 | 0.3 | 0.5 | 0.5 | 0.5 | 1.0 | 1.0 | 1.0 |
| Hallucination in generative responses: RAG systems still face severe hallucination when generating answers, which cannot be completely avoided even with image extraction and structured queries. | 0.5 | 0.5 | 0.5 | 0.5 | 0.5 | 0.8 | 0.8 | 0.8 |

### 4.3.2 Methodological Validation and Application Potential

The experimental results demonstrate the significant value of the paradigm in analyzing technical breakthroughs within the RAG domain. This validation fully proves the paradigm's application potential in broader scientific fields. From biomedicine to materials science, from social sciences to engineering technology, every field has similar patterns of technological evolution and breakthroughs. The paradigm can provide systematic research trend analysis and decision support for these fields, promoting the intelligent transformation of scientific research paradigms.

## 5 Discussion

### 5.1 Academic Contributions and Innovations

This study achieves three core innovations: the multi-stage knowledge construction process effectively solves issues of knowledge fragmentation and incompleteness; the temporal evolution analysis mechanism provides a new method for tracking research trends; and the multi-agent collaboration framework enables intelligent decomposition and execution of research tasks.

## 5.2 Application Value and Limitations

The paradigm can automate literature review and trend analysis, providing data-driven support for research decisions. Through structured knowledge representation and traceable reasoning processes, it enhances the transparency and verifiability of scientific processes.

The main limitations include: high processing speed and resource consumption; insufficient capability in creative thinking and disruptive innovation; and current focus on computer science domains, with applicability in other disciplines requiring further validation. Future work will focus on optimizing algorithm efficiency, expanding application domains, enhancing creative reasoning capabilities, and deepening human-machine collaboration mechanisms.

# 6 Conclusion

This paper proposes a novel self-automated research paradigm based on dynamic knowledge graphs and multi-agent collaboration. Through innovative mechanisms such as multi-stage knowledge construction, temporal evolution analysis, and multi-hop reasoning, the paradigm achieves end-to-end intelligent support from knowledge discovery to paper generation.

The experimental validation in the RAG technology domain demonstrates that the paradigm can automatically track the evolution patterns of research challenges and generate high-quality research analysis reports. This successful validation confirms the paradigm's significant potential for broader application in scientific fields, laying an important methodological foundation for the development of the "Agentic Science" paradigm and driving the intelligent transformation of scientific research.

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
