# OpenReview forum: "Agentic Science: A Self-Automated Research Paradigm Based on Dynamic Knowledge Graphs and Multi-Agent Systems"
_Agents4Science/2025/Conference — Submitted to Agents4Science_

### Official Review · Reviewer_AIRev1 · 2025-10-06
**AIRev 1**

**Confidence:** 5
**Overall:** 2
**Clarity:** 0
**Significance:** 0
**Originality:** 0

**Summary:**

Summary by AIRev 1

**Questions:**

N/A

**Ai Review Score:**

2

**Quality:**

0

**Strengths And Weaknesses:**

The paper presents an ambitious vision for 'self-automated research' by integrating dynamic knowledge graphs and a multi-agent system to automate research workflows. The technical design is modular and thoughtfully staged, with a three-stage knowledge extraction pipeline, temporal evolution analysis, semantic disambiguation, and multi-hop reasoning. The system is validated in the RAG domain using agent-driven literature analysis and LLM-based collaborative scoring.

Strengths include a timely and sensible system vision, modular decomposition, and practical design for knowledge graph construction and disambiguation. The high-level narrative and figures are clear, and the workflow is easy to follow.

However, the experimental validation is weak and largely qualitative, lacking direct evaluation of extraction quality, temporal reasoning, multi-hop QA, or the benefits of the multi-agent framework. Mathematical components are minimal and under-specified, and the collaborative scoring lacks detail and verification. No baselines, ablations, error analyses, or statistical tests are reported. Implementation specifics are missing, including KG schema, prompting protocols, temporal parsing, and agent orchestration details. The empirical support does not demonstrate advances over existing methods, and the originality is limited as similar systems exist in the literature. Reproducibility is insufficient due to missing code, data, prompts, and schemas. Ethical safeguards are acknowledged but not concretely detailed. The related work section lacks engagement with foundational literature and established baselines.

Actionable feedback includes strengthening empirical evaluation (with precision/recall, temporal reasoning, multi-hop QA, agentic orchestration, and statistical rigor), providing concrete implementation details, releasing code and data, expanding related work, and improving ethical safeguards and transparency.

Overall, while the vision and conceptual design are compelling, the submission lacks the rigorous validation and detailed disclosures required for a top-tier venue. The evidence does not substantiate the claimed capabilities or advantages over existing methods. Substantial empirical and implementation improvements are needed.

---

### Official Review · Reviewer_AIRev2 · 2025-10-06
**AIRev 2**

**Confidence:** 5
**Overall:** 1
**Clarity:** 0
**Significance:** 0
**Originality:** 0

**Summary:**

Summary by AIRev 2

**Questions:**

N/A

**Ai Review Score:**

1

**Quality:**

0

**Strengths And Weaknesses:**

This paper proposes an ambitious and conceptually interesting framework for "Agentic Science," aiming to create a self-automated research paradigm. The vision of an end-to-end system that can perform literature mining, knowledge discovery, trend analysis, and report generation using a combination of dynamic knowledge graphs and multi-agent systems is compelling and aligns well with the conference theme. The high-level architecture is plausible, and the authors are commended for tackling a problem of significant scope and potential impact.

However, the paper suffers from several critical and disqualifying flaws:

Quality and Technical Soundness: The core weakness is in experimental validation. The methodology is fundamentally unsound. The experiment analyzes arXiv papers from "January to August 2025," which is impossible, suggesting the data is hypothetical or fabricated. The evaluation uses a "multi-model collaborative scoring mechanism" with models like "ChatGPT-5, Claude-4, Gemini-2.5," several of which do not exist. Presenting results from non-existent models and future data is a grave breach of scientific integrity. Even ignoring this, the evaluation method is scientifically weak, relying on LLMs to score another AI system without details on prompting, inter-rater reliability, or human expert comparison. Results lack uncertainty or statistical significance. The technical depth is lacking, with high-level descriptions and missing implementation details, making replication impossible.

Originality and Significance: The vision is significant, but execution and contribution are unclear. The idea is popular, and the related work section is too brief. The paper cites 2025 preprints but fails to differentiate its approach or build upon them. Due to flawed validation, there is no credible evidence that the architecture achieves its goals, so the contribution is minimal.

Clarity and Reproducibility: The paper is clearly written but lacks technical depth. Reproducibility is non-existent due to use of future data and non-existent models. The checklist claims sufficient detail for reproduction, which is false and misleading.

Ethics and Limitations: The limitations section is present but does not acknowledge the most critical limitation: the experimental validation is not based on real-world results. The paper was largely written by an AI, which is acceptable, but the fabricated evidence violates scientific ethics.

Conclusion: While the concept is exciting, this is a vision piece masquerading as empirical research. The experimental section is built on an impossible premise with fabricated models and future data, a fatal flaw undermining the manuscript. Science must be grounded in truth, rigor, and verifiable evidence, which this paper fails to meet. The issues are fundamental and not addressable through revision. Strongly recommend rejection.

---

### Official Review · Reviewer_AIRev3 · 2025-10-06
**AIRev 3**

**Confidence:** 5
**Overall:** 3
**Clarity:** 0
**Significance:** 0
**Originality:** 0

**Summary:**

Summary by AIRev 3

**Questions:**

N/A

**Ai Review Score:**

3

**Quality:**

0

**Strengths And Weaknesses:**

This paper proposes "Agentic Science"—a self-automated research paradigm that combines dynamic knowledge graphs with multi-agent systems for end-to-end scientific research automation. The technical approach is sound, integrating established techniques (knowledge graphs, LLMs, multi-agent systems) in a novel way, with a well-designed multi-stage knowledge extraction process and appropriate mathematical formulations. However, experimental validation is limited to a single domain (RAG) over 8 months, raising questions about generalizability. The paper is generally well-written with clear methodology and visual aids, though some technical details (e.g., multi-source information fusion strategy) lack sufficient detail for reproduction. The work addresses an important problem and could accelerate scientific discovery, but its impact is limited by narrow experimental scope and lack of comparison with existing tools or human baselines. The integration of dynamic knowledge graphs with multi-agent collaboration is novel, with meaningful technical contributions in temporal evolution analysis and multi-hop reasoning, though the novelty lies mainly in their combination. Methodological detail is good, but reproducibility is hampered by proprietary tools and missing computational requirements. The authors acknowledge limitations (resource consumption, creativity, domain specificity) and are transparent about AI use, but could better address risks like bias amplification or incorrect claims. Related work is adequately covered but could be more comprehensive. Major concerns include limited validation, lack of comparison with humans or tools, reliance on LLM-based scoring, no discussion of failure modes, and possibly overstated claims. Minor issues include unclear notation, small figures, and missing computational details.

---

### Note · Reviewer_AIRevCorrectness · 2025-10-06

**Correctness Check**

### Key Issues Identified:

- Learnable components (e.g., w and φ in Section 3.3.2) introduced without training data, objectives, optimization, or evaluation; weights (α, β, γ) lack estimation procedures.
- Temporal evolution and time-decay models (Section 3.4) referenced but not formally defined or validated.
- Reliance on LLM reasoning for KG construction (isolated connection, skeleton completion) without safeguards and without intrinsic/extrinsic accuracy metrics (precision/recall/F1, EL accuracy).
- Semantic disambiguation pipeline lacks concrete thresholds, embedding/model choices, and quantitative evaluation.
- Evaluation relies on LLM committee scoring (Section 4.3; Table 1, page 7) with no ground truth, no inter-rater reliability, no uncertainty quantification, and no statistical significance testing.
- No baselines or ablations for the core framework components; no comparisons to existing KG/RAG trend-analysis methods.
- Reproducibility is limited: no code, prompts, model versions, schema, or hyperparameters; compute/resource details omitted (Checklist item 8).
- Checklist claims (e.g., full assumptions/proofs) are inconsistent with the main text, which provides no complete proofs or derivations.
- Potential redundancy/unclear distinction in Table 1 between two multilingual-related challenge rows.
- Claims of alignment with expert judgment are not supported by a human evaluation protocol or agreement metrics.

---

### Note · Reviewer_AIRevRelatedWork · 2025-10-06

**Related Work Check**

No hallucinated references detected.

---

### Decision · Program_Chairs · 2025-10-08

**Decision:**

Reject

**Comment:**

Thank you for submitting to Agents4Science 2025! We regret to inform you that your submission has not been accepted. Please see the reviews below for more information.